# Influence of O_2_ Flow Rate on the Properties of Ga_2_O_3_ Growth by RF Magnetron Sputtering

**DOI:** 10.3390/mi14020260

**Published:** 2023-01-19

**Authors:** Dengyue Li, Hehui Sun, Tong Liu, Hongyan Jin, Zhenghao Li, Yaxin Liu, Donghao Liu, Dongbo Wang

**Affiliations:** 1CNPC Bohai Drilling Engineering Company Limited, Tuanjie Eeat Road, Haibin Street, Binhai Area, Tianjin 300457, China; 2Chinese Academy of Sciences, Suzhou Institute Nanotech & Nanobionics SINANO, Vacuum Interconnected Nanotech Workstat Nano X, Suzhou 215123, China; 3Suzhou Institute of Product Quality Supervision and Inspection, Suzhou 215128, China; 4National Key Laboratory for Precision Hot Processing of Metals, Harbin Institute of Technology, Harbin 150001, China; 5Department of Optoelectronic Information Science, School of Materials Science and Engineering, Harbin Institute of Technology, Harbin 150001, China

**Keywords:** Ga_2_O_3_, O_2_ flow rate, magnetron sputtering

## Abstract

The influence of the O_2_ flow rate on the properties of gallium oxide (Ga_2_O_3_) by RF magnetron sputtering was studied. X-ray diffraction (XRD), atomic force microscopy (AFM), scanning electron microscopy (SEM), transmittance spectra, and photoluminescence (PL) spectra have been employed to study the Ga_2_O_3_ thin films. With the increase in oxygen flow rate, both the crystal quality and luminescence intensity of the Ga_2_O_3_ samples first decrease and then enhance. All these observations suggested that the reduction in the oxygen defect density is responsible for the improvement in the crystal quality and emission intensity of the material. Our results demonstrated that high-quality Ga_2_O_3_ materials could be obtained by adjusting the oxygen flow rate.

## 1. Introduction

Recently, gallium oxide (Ga_2_O_3_) and related compounds (Al_x_Ga_2-X_O_3_, (In_x_Ga_1-x_)_2_O_3_) [1,2] have brought about the widespread attention, attributed to their outstanding electrical and photoelectric properties, such as wide fundamental bandgap (4.5–5 eV), a high off-state breakdown voltage of 755 V, high dielectric constant values from 10.2 to 14.2, and high mobility of 2790 cm^2^ V^−1^ s^−1^ [3,4,5]. At present, Ga_2_O_3_ has been widely used in solar blind ultraviolet detection [6], high power switching [7], metal oxide semiconductor field effect transistors (MOSFET) [8], high-temperature gas sensors [9], and many other fields.

Ga_2_O_3_ usually exists in six different polymorphic structures (α, β, γ, δ, ε, and k) [10]. Among these, β-Ga_2_O_3_ is considered as the most stable phase and can be converted from other phases at high temperatures [11,12]. So far, many growth modes were used to develop β-Ga_2_O_3_, including RF Sputter [13], MBE [14], MOCVD [15], chemical vapor deposition [16], etc. RF magnetron sputtering is a comparatively economical deposition technique that has adequate control over stoichiometry and uniformity of the film compared to the above techniques. Until now, the effects of growth parameters, for instance, substrate temperature, oxygen/argon partial pressures, and sputtering power, on the properties of Ga_2_O_3_ have been studied the most [17,18,19]. However, hardly any reports about the effect of the O_2_ flow rate at fixed Ar flow rate on the structure and optical properties of β-Ga_2_O_3_ thin films deposited by the reactive RF magnetron sputter. Since oxygen deficiency in the growth process can induce oxygen vacancies in Oxide semiconductor materials, oxygen vacancies affect the optical and electrical properties of oxide semiconductor films [20,21,22,23,24]. Therefore, it is of great significance to study the effect of the O_2_ flow rate on the characteristics of Ga_2_O_3_ thin films deposited by RF magnetron sputtering.

In this work, β-Ga_2_O_3_ has been grown by RF magnetron sputtering, and the effect of O_2_ flow rate on the structure and optical characteristic of β-Ga_2_O_3_ have been studied in detail. The improvement of UV emission properties was observed in the β-Ga_2_O_3_ samples with an increased O_2_ flow rate. The mechanism of enhanced luminescence in the β-Ga_2_O_3_ film was an in-depth study by careful inspection of the PL spectrum combined with XRD results. It is anticipated that this work will provide a meaningful step toward the fabrication of high-quality β-Ga_2_O_3_ thin films.

## 2. Materials and Methods

Ga_2_O_3_ samples were grown on single-polished c-plane (0006) sapphire substrates using an RF magnetron sputtering system. A sintered ceramic Ga_2_O_3_ target of 99.99% purity was employed as the target. Before growth, the sapphire substrates were first cleaned with ultrasonic vibration in ethanol and then in high purity water. The argon gas flow rate was set at 30 sccm and pressure at 0.8 Pa. Sputtering power was adjusted to 80 w. The distance between the sample and target was 760 mm. The base pressure of vacuum chamber reached 5.6 × 10^−4^ Pa. The Ar flow rate was kept constant at 30 sccm. In the experiments, the O_2_ flow rate was set as 0 sccm, 1 sccm, 2 sccm, 4 sccm, respectively. The influence of other parameters was minimized.

The structure of Ga_2_O_3_ was characterized by an XRD technique (XRD, X’ Pert, Philips, Eindhoven, The Netherlands). The morphologies of samples were conducted on a field-emission scanning electron microscopy (FE-SEM, ZEISS Merlin Compact, Oberkochen, Germany). Photoluminescence (PL) spectrums were investigated by Zolix responsivity measurement system (λ = 266 nm) as the excitation source (DSR600, Zolix, Beijing, China). 

## 3. Results

Figure 1 shows the result of the X-ray diffraction of the Ga_2_O_3_ films growth with various O_2_ flow rates. The diffraction peaks located at 29.7°, 37.6°, and 58.4° originate from the 400, 402, and 603 of the β-Ga_2_O_3_, respectively [1,25,26]. For the sample without the O_2_ flow rate, 400, 402, and 603 of the β-Ga_2_O_3_ diffraction peak coexisted; this suggests that the sample was polycrystalline. With the O_2_ flow rate increased from 0 to 4 sccm, the diffraction peak intensity of the 400 β-Ga_2_O_3_ decreased, while the intensity of both the 402 and 603 of β-Ga_2_O_3_ diffraction peak increased. Both of these two diffractions belong to the 201 plane family of the monoclinic Ga_2_O_3_ [27,28]. The above result illustrates that highly 201-textured β-Ga_2_O_3_ samples have been prepared and the orientation of crystal is gradually enhanced when oxygen flow increased. Furthermore, the full width at half maximum (FWHM) values of the 402 β-Ga_2_O_3_ peaks are 1.00°, 1.10°, 1.06°, and 0.96° for samples with the O_2_ flow rate increased from 0 to 4 sccm, respectively. The FWHM value is dependent on the O_2_ flow rate, and the results suggest a higher O_2_ flow rate results in improved crystal quality. The minimal FWHM is obtained at 4 sccm of the O_2_ flow rate, which means the grain size is the largest [29]. The combined results of the XRD peak intensity and the FWHM value of the samples show that higher O_2_ flow rates lead to better quality.

Figure 2 shows AFM images of the samples. The root mean square (RMS) roughness is 0.606 nm, 8.23 nm, 3.41 nm, and 1.23 nm for samples with different O_2_ flow rates, respectively. The RMS value identified with the XRD results indicates that the appropriate O_2_ flow rate gives rise to the improvement of the Ga_2_O_3_ structure.

The EDX spectroscopy analyses of the Ga_2_O_3_ films is shown in Figure 3. In the graph, O and Ga peaks can be observed. The composition of the Ga_2_O_3_ thin films are shown in Table 1. The atomic concentration of the O composition decreases from 39.88 to 10 at% with the rise in O_2_ flow rate. However, as the O_2_ flow rate continues to increase, the oxygen content starts to rise again.

The thickness of samples measured by cross section scanning electron microscopy (Figure 4) is also given in Table 1. According to the data in the table, the thickness of the sample decreases gradually as the O_2_ flow rate increases.

Combined with XRD and AFM results, the Ar partial pressure in the cavity going down, the target atoms produced by bombardment decreasing, and both the growth rate and crystallinity of the Ga_2_O_3_ sample decreasing can be attributed to the O_2_ flow rate beginning to rise. As the O_2_ flow continues to rise, the oxygen vacancy defects decrease, and the crystallinity of gallium oxide films is improved.

The transmission spectrum of the sample is shown in Figure 5. All the sample’s transmissibility is over 75% and has interference fringes, indicating the existence of a smooth surface. As the O_2_ flow increases from 0 to 1 sccm, the transmittance of the sample decreases rapidly. With the increase in oxygen flow from 1 to 4 sccm, the transmittance of the sample increases gradually. The results of transmission spectrum and crystal mass can confirm each other; with the rise of O_2_ flow rates, the crystal mass of the sample decreases first and then increases, and the transmission spectrum also shows the same rule. In addition, the band gap of the sample becomes widened when the oxygen flow increases from 1 to 4 sccm.

The PL spectra of Ga_2_O_3_ in the UV region at room temperature is shown in Figure 6a. The emission peak at 266 nm (4.66 eV) originated from the Ga_2_O_3_ samples [4,11]. When the O_2_ flow rate increased from 0 to 1 sccm, the intensity of Ga_2_O_3_ emission peaks decreased, and as the oxygen flow continued to rise, the intensity of Ga_2_O_3_ emission peaks increased. Combined with the above analysis of crystal quality, the PL result can be interpreted as when the O_2_ flow rate increased from 0 to 1 sccm, the crystalline quality of the sample deteriorated, which caused decreases in the luminescence intensity. As the oxygen flow continues to rise, the oxygen defect density decreases and the non-radiative composite center decreases, and this ultimately causes the luminescence intensity to increase. This is due to oxygen deficiency in the growth process producing oxygen defects in Ga_2_O_3_ and oxygen defects playing the role of non-radiation complex centers, and thus as oxygen flow continue to rise, the number of oxygen defects decreases and this increases the intensity of the Ga_2_O_3_ emission peaks.

To confirm the discussion above, Figure 6b shows the PL spectra samples in the visible region. The emission band in the region of 450–600 nm in all PL spectra can be attributed to oxygen-defect-related deep-level emission [30,31,32,33]. It can be seen that the intensity of the oxygen-defect-related emission peak decreases gradually with the increase in the oxygen flow rate. Therefore, it is reasonable to conclude that increasing the oxygen flow rate leads to reductions in the oxygen defect density and improvements in the crystal quality and emission intensity of the material.

## 4. Conclusions

In summary, in terms of the effect of oxygen flow on the structure, optical l properties of the Ga_2_O_3_ films have been investigated by XRD, EDX, AFM, transmission spectra, and PL spectra. With the increase in the oxygen flow rate, both the crystal quality and luminescence intensity of the sample first decreased and then enhanced. All these observations suggested that the reduction in the oxygen defect density is responsible for the improvement in the crystal quality and emission intensity of the material, however, there have been no reports about O_2_ flow rate on the properties of the Ga_2_O_3_ growth by RF magnetron sputtering. Our results were similar to those obtained by other techniques and the specific control of various experimental operating parameters. Vu found that the performance of β-Ga_2_O_3_-based photodetectors with a higher oxygen partial are better than those prepared at lower oxygen pressures [34]. Wang et al. studied the influence of oxygen flow ratio on the performance of Sn-doped Ga_2_O_3_ films by RF magnetron sputtering; they found the sample with higher oxygen flow ratio displays an enhanced performance [35]. Shen’s study revealed oxygen annealing will enhance the performance of β-Ga_2_O_3_ solar-blind photodetectors grown by ion-cutting process [36]. Our results demonstrated that high-quality gallium oxide materials can be obtained by adjusting the oxygen flow rate.

## Figures and Tables

**Figure 1 micromachines-14-00260-f001:**
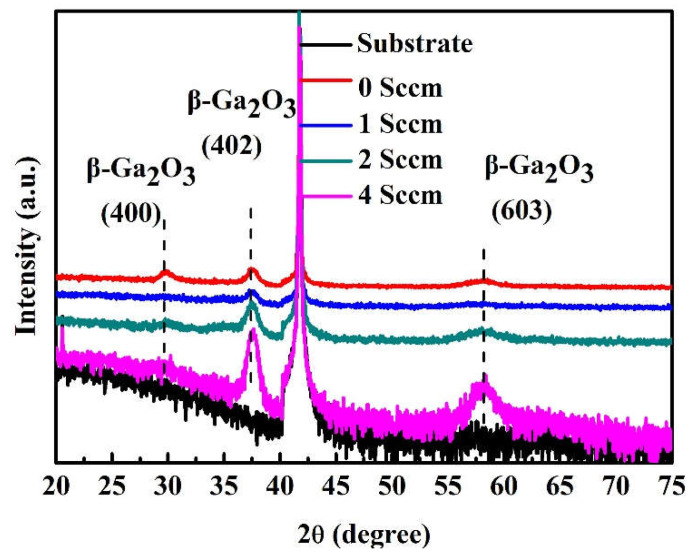
XRD patterns of Ga_2_O_3_ films with various O_2_ flow rates.

**Figure 2 micromachines-14-00260-f002:**
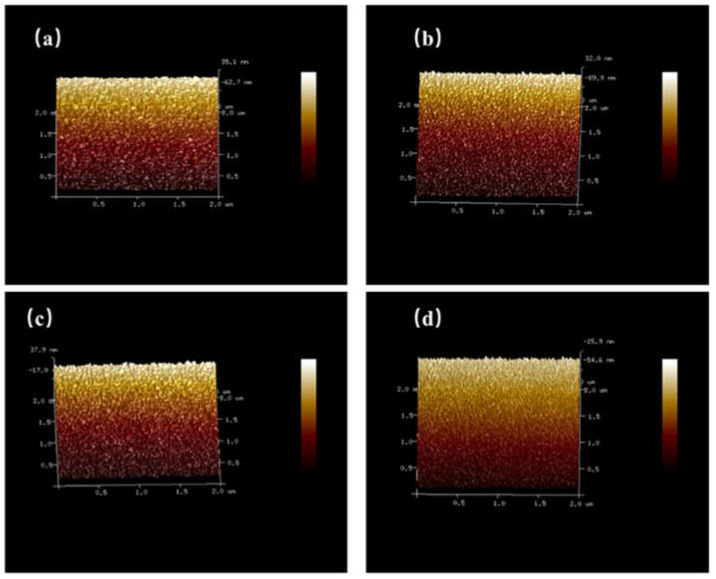
AFM images of Ga_2_O_3_ films with various O_2_ flow rates (5 × 5 µm^2^). (**a**). 0sccm; (**b**). 1sccm; (**c**). 2 sccm; (**d**). 4sccm.

**Figure 3 micromachines-14-00260-f003:**
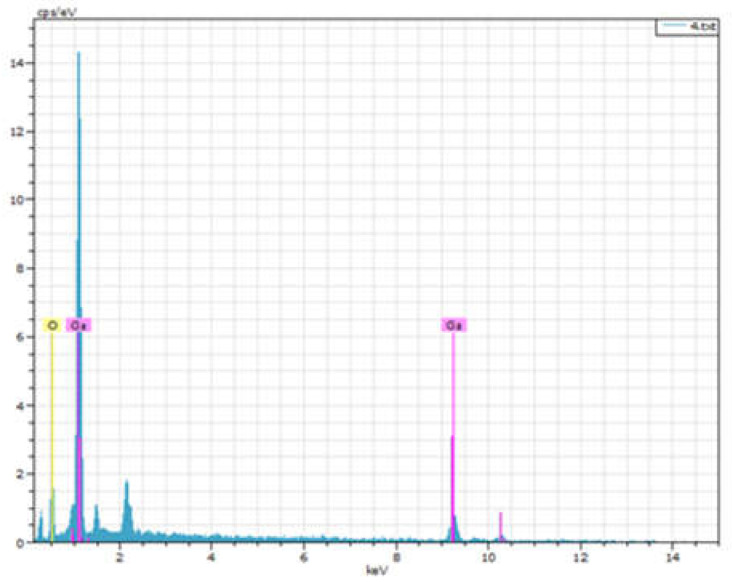
EDX spectroscopy analyses of the sample.

**Figure 4 micromachines-14-00260-f004:**
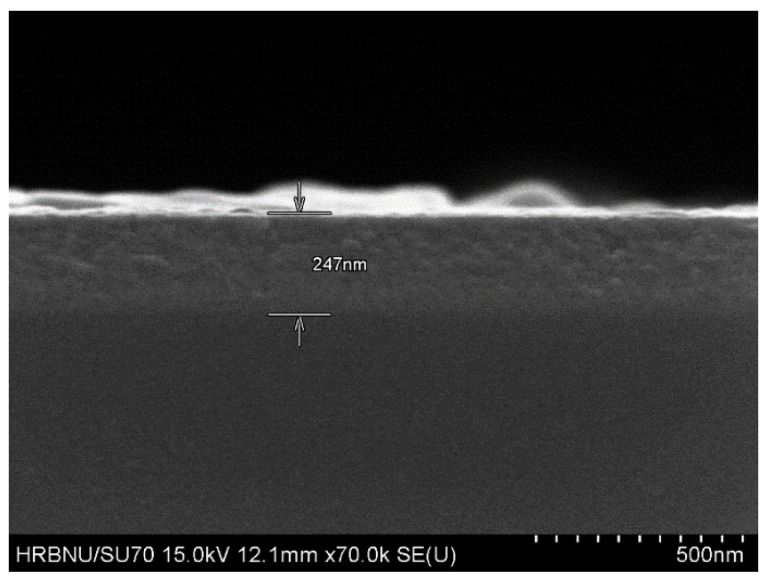
Cross section scanning of the Ga_2_O_3_.

**Figure 5 micromachines-14-00260-f005:**
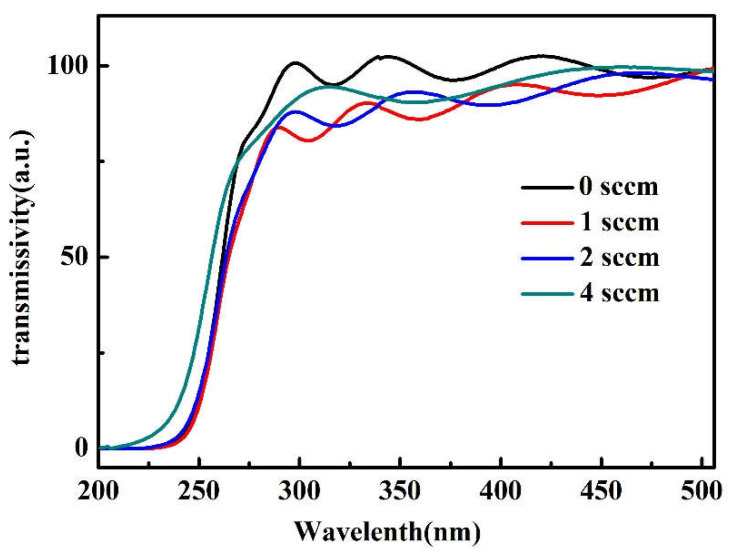
Transmission spectrum of the Ga_2_O_3_.

**Figure 6 micromachines-14-00260-f006:**
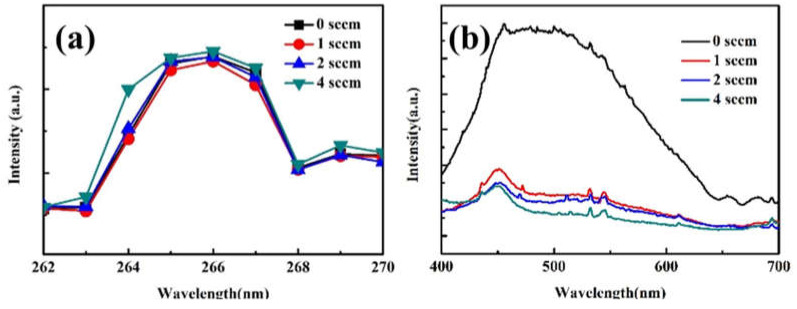
PL spectra of Ga_2_O_3_ samples at room temperature. (**a**). UV region; (**b**). Visible region.

**Table 1 micromachines-14-00260-t001:** Oxygen content and thickness of samples.

O_2_ Flow Rate (sccm)	O Atom [at %]	Thickness
0	39.88%	350 nm
1	10.00%	330 nm
2	19.03%	300 nm
4	47.62%	247 nm

## Data Availability

Data is readily available, when someone asks for it.

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
