# Peer review of "Influence of O2 Flow Rate on the Properties of Ga2O3 Growth by RF Magnetron Sputtering"

_micromachines, 2023, doi:10.3390/mi14020260_

Round 1

Reviewer 1 Report

The paper is devoted to the study of the effect of oxygen flow rate on the characteristics of gallium oxide thin films deposited by RF magnetron  sputtering. In general the methodology is given clearly and the conclusions are supported by the data. The presentation of the paper, however, leaves some room for improvement. The English language of the paper needs to be improved, as some sentences are difficult to understand (such as 'A Ga2O3 target was prepared by a sintering mixture of 99.99 % pure Ga2O3 powders and 60.0 mm diameter.' In the abstract, a method of synthesis should be specified explicitly. Reference list should be re-considered, (Ref. 1 is missing, there are Refs. 1 and 2 after Ref. 25, etc.). Also, oxygen defects are indeed known to affect the properties of gallium oxide very strongly. How do the authors' data relate to similar results reported in the literature? After these issues are addressed, the paper, in the Reviewer's opinion, can be published in Micromachines.  

Author Response

Response letter

Dear Editor and reviewers,

  Thank you very much for handling our manuscript, and we also appreciate the reviewers for their good suggestions and valuable comments. We have made the revisions based on the comments. Please see our detailed point-to-point response as follows. All changes made accordingly in the manuscript are highlighted in red. We hope all your concerns have been well addressed, and the quality of our paper has been greatly improved after the revisions.

REVIEWER COMMENTS

Reviewer #1:

The paper is devoted to the study of the effect of oxygen flow rate on the characteristics of gallium oxide thin films deposited by RF magnetron sputtering. In general the methodology is given clearly and the conclusions are supported by the data. The presentation of the paper, however, leaves some room for improvement. The English language of the paper needs to be improved, as some sentences are difficult to understand (such as 'A Ga2O3 target was prepared by a sintering mixture of 99.99 % pure Ga2O3 powders and 60.0 mm diameter.' In the abstract, a method of synthesis should be specified explicitly. Reference list should be re-considered, (Ref. 1 is missing, there are Refs. 1 and 2 after Ref. 25, etc.). Also, oxygen defects are indeed known to affect the properties of gallium oxide very strongly. How do the authors' data relate to similar results reported in the literature? After these issues are addressed, the paper, in the Reviewer's opinion, can be published in Micromachines.

[Overall response]

Thank you very much for your kind review and your suggestions to improve our manuscript. We have revised the manuscript accordingly.

Comment 1. The English language of the paper needs to be improved, as some sentences are difficult to understand (such as 'A Ga2O3 target was prepared by a sintering mixture of 99.99 % pure Ga2O3 powders and 60.0 mm diameter.' In the abstract, a method of synthesis should be specified explicitly. Reference list should be re-considered, (Ref. 1 is missing, there are Refs. 1 and 2 after Ref. 25, etc.).

[Response] Thank you for your comments. We have learned a lot.

We have revised the article, the method of synthesis is specified explicitly, and Reference list re-considered.

Comment 2. Also, oxygen defects are indeed known to affect the properties of gallium oxide very strongly. How do the authors' data relate to similar results reported in the literature?

[Response] Thank you very much for your question.

Although there have been no reports about O2 flow rate on the properties of Ga2O3 growth by RF magnetron sputtering. Our results were similar to those obtained by other techniques and specific control of various experimental operating parameters. Vu found the performance of β-Ga2O3-based photodetector with a higher oxygen partial are better than those prepared at lower oxygen pressures [37]. Wang et al studied. The Influence of oxygen flow ratio on the performance of Sn-doped Ga2O3 films by RF magnetron sputtered, they found sample with higher oxygen flow ratio displays an enhanced performance [38]. Shen’s study revealed oxygen annealing will enhance the performance of β-Ga2O3 solar-blind photodetectors grow by ion-cutting process [39].

Please feel free to contact us with any questions and we are looking forward to your consideration.

Best regards.

Yours sincerely,

Dongbo Wang

Email: wangdongbo@hit.edu.cn

Reviewer 2 Report

11      It is mentioned that the structural and optical properties of β-Ga2O3 films depending on the O2 flow rate have been investigated for the first time and results have been supported by XRD, EDX, AFM, transmission spectra and PL spectra.

2     Selection of β-Ga2O3 under investigation is justified by the authors.

3    All the experimental verification of different structural and optical properties are confirmed with all types of experimental techniques like XRD, EDX, AFM, transmission spectra and PL spectra.

4    There seems to be pointless justification for going for all these types of experimental techniques just to confirm the outcome.

5    The final result that the reduction of the oxygen defect density is responsible for the improve the crystal quality and emission intensity of the material can still play a second control to other parameters like Argon flow, flow rate, time, and many more, so it will be hypothetical to reach to any such definite conclusion about the dependency of properties on oxygen flow rate alone.

6     No theoretical basis has also been given to strengthen the experimental results.

7    RF sputtering method is considered where as it may have disadvantages/limitations as compared to other techniques of synthesizing β-Ga2O3 films.

8    How all the sputtering parameters are said to be controlled to fixed values and only oxygen flow rate is isolated.

9    With the increase of oxygen flow rate, both the crystal quality and luminescence intensity of the Ga2O3 samples are said to be first decreased and then enhanced.

10.  Outcomes of samples are said to be with improved quality of texture, increased FWHM values (i.e., improvement of crystallinity) but these outcomes needs to compared with other techniques and specific control of various experimental operating parameters before reaching to some definite conclusions.

11.  English of the manuscript also needs improvement.  

Author Response

Response letter

Dear Editor and reviewers,

  Thank you very much for handling our manuscript, and we also appreciate the reviewers for their good suggestions and valuable comments. We have made the revisions based on the comments. Please see our detailed point-to-point response as follows. All changes made accordingly in the manuscript are highlighted in red. We hope all your concerns have been well addressed, and the quality of our paper has been greatly improved after the revisions.

REVIEWER COMMENTS

Reviewer #2:   

[Overall response] We sincerely appreciate your kind review and suggestions to improve our manuscript.

Comment 1. It is mentioned that the structural and optical properties of β-Ga2O3 films depending on the O2 flow rate have been investigated for the first time and results have been supported by XRD, EDX, AFM, transmission spectra and PL spectra.

[Response] Thank you for your careful review.

Comment 2.  Selection of β-Ga2O3 under investigation is justified by the authors.

[Response] Thank you for your careful review.  

Comment 3. All the experimental verification of different structural and optical properties are confirmed with all types of experimental techniques like XRD, EDX, AFM, transmission spectra and PL spectra.

[Response] Thank you for your careful review.

Comment 4. There seems to be pointless justification for going for all these types of experimental techniques just to confirm the outcome.

[Response] Thank you for your suggestions. We are grateful for the suggestion.

Ga2O3 can form six different polymorphic structures, known as α, β, γ, δ, ε, and k. Among these, β-Ga2O3 is regarded as the most stable phase that could be converted from other metastable phases at high temperatures. So far, many growth methods have been used to grow β-Ga2O3, including RF Sputter, MBE, MOCVD, chemical vapor deposition, etc. RF magnetron sputtering a relatively cost-effective deposition technique compared with those listed above, has sufficient control over the stoichiometry and uniformity of the thin films. Until now, the physical properties of β-Ga2O3 films depend on sputtering parameters, such as substrate temperature, oxygen/argon partial pressures and sputtering power have been reported.

However, reports about the effect of oxygen flow rate at constant Ar flow rate on the characteristics of reactive RF magnetron sputter deposited β-Ga2O3 thin films are limited. Since oxygen deficiency in the growth process can induce oxygen vacancies in Oxide semiconductor material, and oxygen vacancies affect the optical and electrical properties of Oxide semiconductor films.

Therefore, it is very significant to study Effect of O2 flow rate on the characteristics of Ga2O3 thin films deposited by RF magnetron sputtering.

Comment 5. The final result that the reduction of the oxygen defect density is responsible for the improve the crystal quality and emission intensity of the material can still play a second control to other parameters like Argon flow, flow rate, time, and many more, so it will be hypothetical to reach to any such definite conclusion about the dependency of properties on oxygen flow rate alone.

[Response] Thank you very much for your question.

Oxygen deficiency in the growth process can induce oxygen vacancies in Oxide semiconductor material, and oxygen vacancies affect the optical and electrical properties of Oxide semiconductor films.

You're right, “the final result that the reduction of the oxygen defect density is responsible for the improve the crystal quality and emission intensity of the material can still play a second control to other parameters like Argon flow, flow rate, time, and many more”

In this letter, we fixed the flow rate of argon gas was set at 30 sccm and pressure at 0.8 Pa. Sputtering power was adjusted to 80 w. The distance between the sample and target was 760 mm. The base pressure of vacuum chamber reached 5.6×10-4 Pa. Minimize the influence of other parameters.

Comment 6. No theoretical basis has also been given to strengthen the experimental results.

[Response] Thank you very much for your question.

We have increased our research on luminescence properties.

Because oxygen deficiency in the growth process will produce oxygen defects in Ga2O3 and oxygen defects play the role of non-radiation complex center, so as oxygen flow continues to rise, the number of oxygen defects decreases, this increases intensity of Ga2O3 emission peaks.

Comment 7. RF sputtering method is considered where as it may have disadvantages/limitations as compared to other techniques of synthesizing β-Ga2O3 films.

[Response] Thank you for your suggestions. We are grateful for the suggestion.

Radio frequency (RF) sputtering is a technique that is used to create thin films, such as those found in the computer and semiconductor industry. Like direct current (DC) sputtering, this technique involves running an energetic wave through an inert gas to create positive ions. The target material, which will ultimately become the thin film coating, is struck by these ions and broken up into a fine spray that covers the substrate, the inner base of the thin film. RF sputtering differs from DC sputtering in the voltage, system pressure, sputter deposition pattern, and ideal type of target material.

During the sputtering process, the target material, substrate, and RF electrodes begin in a vacuum chamber. Next, the inert gas, which is usually argon, neon, or krypton, depending on the size of the target material’s molecules, is directed into the chamber. The RF power source is then turned on, sending radio waves through the plasma to ionize the gas atoms. Once the ions begin to contact the target material, it is broken into small pieces that travel to the substrate and begin to form a coating.

Radio-frequency magnetron sputtering deposition, which has the merits of easy controllability, high efficiency, strong adhesion, and low cost, has been used to fabricate various films. Easily prepared targets such as ceramics, metals, and oxides are convenient for use in magnetron sputtering to deposit films.

Sputtering technique have disadvantages of small grain size and rough surface, and so on.

Comment 8. How all the sputtering parameters are said to be controlled to fixed values and only oxygen flow rate is isolated.

[Response] Thank you for your suggestions. We are grateful for the suggestion.

In this letter, we fixed the flow rate of argon gas was set at 30 sccm and pressure at 0.8 Pa. Sputtering power was adjusted to 80 w. The distance between the sample and target was 760 mm. The base pressure of vacuum chamber reached 5.6×10-4 Pa. Minimize the influence of other parameters.

Comment 9.    With the increase of oxygen flow rate, both the crystal quality and luminescence intensity of the Ga2O3 samples are said to be first decreased and then enhanced.

[Response] Thank you for your careful review.

Comment 10.  Outcomes of samples are said to be with improved quality of texture, increased FWHM values (i.e., improvement of crystallinity) but these outcomes needs to compared with other techniques and specific control of various experimental operating parameters before reaching to some definite conclusions.

[Response] Thank you for your suggestions. We are grateful for the suggestion.

Although there have been no reports about O2 flow rate on the properties of Ga2O3 growth by RF magnetron sputtering. Our results were similar to those obtained by other techniques and specific control of various experimental operating parameters. Vu found the performance of β-Ga2O3-based photodetector with a higher oxygen partial are better than those prepared at lower oxygen pressures [37]. Wang et al studied. The Influence of oxygen flow ratio on the performance of Sn-doped Ga2O3 films by RF magnetron sputtered, they found sample with higher oxygen flow ratio displays an enhanced performance [38]. Shen’s study revealed oxygen annealing will enhance the performance of β-Ga2O3 solar-blind photodetectors grow by ion-cutting process [39].

Comment 11.  English of the manuscript also needs improvement. 

[Response] Thank you for your suggestions. We are grateful for the suggestion.

We have revised the article

Please feel free to contact us with any questions and we are looking forward to your consideration.

Best regards.

Yours sincerely,

Dongbo Wang

Email: wangdongbo@hit.edu.cn
